# Key learnings from an outcome and embedded process evaluation of a direct to beneficiary mobile health intervention among marginalised women in rural Bihar, India

Laili Irani [1], Supriya Verma,[2] Ruchika Mathur,[2] Raj Kumar Verma,[2] Diwakar Mohan,[3] Diva Dhar,[4] Aaditeshwar Seth,[5] Indrajit Chaudhuri,[6] Mahua Roy Chaudhury,[7] Apolo Purthy,[7] Ankit Nanda,[2] Shivani Singh,[2] Akshay Gupta,[5] Amnesty Elizabeth LeFevre [8]

For numbered affiliations see end of article.

**Correspondence to**
Dr Laili Irani; Laili.ir@gmail.com

## ABSTRACT

**Introduction** Mobile Vaani was implemented as a pilot programme across six blocks of Nalanda district in Bihar state, India to increase knowledge of rural women who were members of self-help groups on proper nutrition for pregnant or lactating mothers and infants, family planning and diarrhoea management. Conveners of self-help group meetings, community mobilisers, introduced women to the intervention by giving them access to interactive voice response informational and motivational content. A mixed methods outcome and embedded process evaluation was commissioned to assess the reach and impact of Mobile Vaani.

**Methods** The outcome evaluation, conducted from January 2017 to November 2018, used a quasi-experimental pre–post design with a sample of 4800 married women aged 15–49 from self-help group households, who had a live birth in the past 24 months. Surveys with community mobilisers followed by meeting observations (n=116), in-depth interviews (n=180) with self-help group members and secondary analyses of system generated data were conducted to assess exposure and perceptions of the intervention.

**Results** From the outcome evaluation, 23% of women interviewed had heard about Mobile Vaani. Women in the intervention arm had significantly higher knowledge than women in the comparison arm for two of seven focus outcomes: knowledge of how to make child's food nutrient and energy dense (treatment-on-treated: 18.8% (95% CI 0.4% to 37.2%, p<0.045)) and awareness of at least two modern spacing family planning methods (treatment-on-treated: 17.6% (95% CI 4.7% to 30.5%, p<0.008)). Women with any awareness of Mobile Vaani were happy with the programme and appreciated the ability to call in and listen to the content.

**Conclusion** Low population awareness and programme exposure are underpinned by broader population level barriers to mobile phone access and use among women and missed opportunities by the programme to improve targeting and programme promotion. Further research is

## STRENGTHS AND LIMITATIONS OF THIS STUDY

⇒ The evaluation used advanced analytical approaches to address low exposure in the outcome evaluation.
⇒ The embedded process evaluation captured the design of how the programme was integrated among self-help group members within a complex socio-cultural environment.
⇒ Since the measurement of exposure was based on reported estimates, future programmes and evaluations should ensure that interactive voice recording call data records are accessible and linkable to survey data.

needed to assess programmatic linkages with changes in health practices.

## INTRODUCTION

When the National Family and Health Survey-4 results came out in 2016–2017, Bihar, India's third largest state (population 99 million), had lower than national averages for most maternal and child health outcomes.[1] In rural Bihar, half the children under 5 and nearly a third (32%) of mothers were underweight as compared with the national figures of 38% and 27%, respectively.[1,2] One in five (21%) married women reported having an unmet need for family planning—an estimate twice that of the national average. The incidence of diarrhoea among children under 5 within the 2 weeks preceding the survey was 11% and less than half of those cases were managed with oral rehydration solution.[2] Against this backdrop, the government of Bihar, with the support of the Bill and Melinda Gates Foundation, established a range of public health

**BMJ**

programmes to improve health systems and bolster health outcomes.

A maternal and child health and nutrition programme was established in 2017, with implementation support provided by Project Concern International.[3] This programme was designed to share health information with women who were members of microfinance-based self-help groups. (A self-help group comprises 8–12 women who live nearby, meet weekly, save small amounts of money to take out loans internally or with banks at small interest rates, and discuss matters related to finance, livelihoods and health.)[3 4] The target audience were mothers of children under 2 years of age who were either self-help group members themselves or were living in a home where at least one other woman residing in the same household was a member of a group. Prior studies had shown that sharing messages on correct maternal and child health and nutrition practices in weekly self-help group meetings coupled with home visits by group leaders and convenors of group meetings, called community mobilisers to women with young children, among others, may improve some maternal and child health outcomes, such as antenatal care visits, early initiation of breast feeding and clean cord care.[4–13] (Community mobilisers are women who support the functioning of ~10 self-help groups within their village. They facilitate weekly meetings for each self-help group.) In an effort to bolster these and other outcomes, including family planning and complementary feeding, the government of Bihar sought to implement and evaluate new and innovative approaches including mobile health solutions which had experienced some success in reinforcing information to target audiences, within selected geographies.[4–8 14–49] Further, as half the state's rural population owned a mobile phone, this new innovative programme held potential.[30 50–52]

Mobile Vaani, developed by Gram Vaani and launched in January 2017, was a direct to beneficiary mobile health communication programme comprising a two-way messaging system which included outbound interactive voice response calls and a mechanism for beneficiaries to call in and record their own content. Trained community mobilisers, with support from programme field officers, promoted awareness of the Mobile Vaani programme in self-help group meetings. At regular intervals, group members were reminded about the programme, and taught how to dial a toll-free number in order to generate a missed call. Within 30 s to 2 min of receiving the missed call, the system would call the number back and if answered, the listener received a series of prerecorded WHO-approved health information content on nutrition,[53] family planning and management of diarrhoea. Call content spanned across four priority health areas (child nutrition, maternal nutrition, family planning and management of diarrhoea) and was comprised of two types of messages: those prerecorded in a studio with actors reading off a script, or messages spontaneously recorded by self-help group women in the same district.

The latter were reviewed and edited before being broadcast. Content was shared through story format which centred around the knowledge and benefits to women, children and families of appropriate maternal and child health and nutrition practices. Listeners were further encouraged to share information about Mobile Vaani with their family members, friends and neighbours, and to foster a dialogue and conversation with others on the topics listened to.

This manuscript presents findings from a mixed methods evaluation conducted with the support of JEEViKA, the government-led entity that establishes and runs self-help groups in Bihar. The evaluation was commissioned as very few interactive voice recording-based programmes had been carried out targeting hard-to-reach marginalised populations and most of these programmes had not been evaluated.[16 22 23] Further, even though Mobile Vaani had been implemented in the general population before, little was known about its reach, the perceptions of the content by its intended audience, or its impact on knowledge for key maternal and child health and nutrition practices among women within self-help group households.[54 55]

Findings on the programme's reach and impact on women's knowledge of key maternal and child health and nutrition indicators captured through the outcome evaluation are shared in this manuscript. It further highlights key results from an embedded process evaluation that helps to explain how the intervention was integrated into the existing system, how the intended beneficiaries engaged with the programme and benefited from it, and how beneficiaries engaged with the technology. Study findings are anticipated to provide important insights into the impact direct to beneficiary solutions have in bolstering knowledge among women.

## METHODS
### Evaluation design and conceptual framework
The mixed methods evaluation design consisted of an outcome evaluation and an embedded process evaluation.

The outcome evaluation used a quasi-experimental before and after study design to identify changes in knowledge on maternal and child nutrition, family planning and management of diarrhoea among self-help group members with a child under 2 years of age. Six intervention blocks were selected by the implementers for programming; six comparison blocks were matched by the evaluators based on the sociodemographic characteristics of female literacy, scheduled caste/schedule tribe and concentration of self-help groups by population.

The *process evaluation* sought to garner in-depth knowledge on the functioning of Mobile Vaani, to assist in interpreting outcome results, and inform potential replication.[27 56–65] Using a realist evaluation approach, the process evaluation helped in building an overall understanding of (1) the system's readiness to accept the intervention, that is, the ability of the community mobilisers to

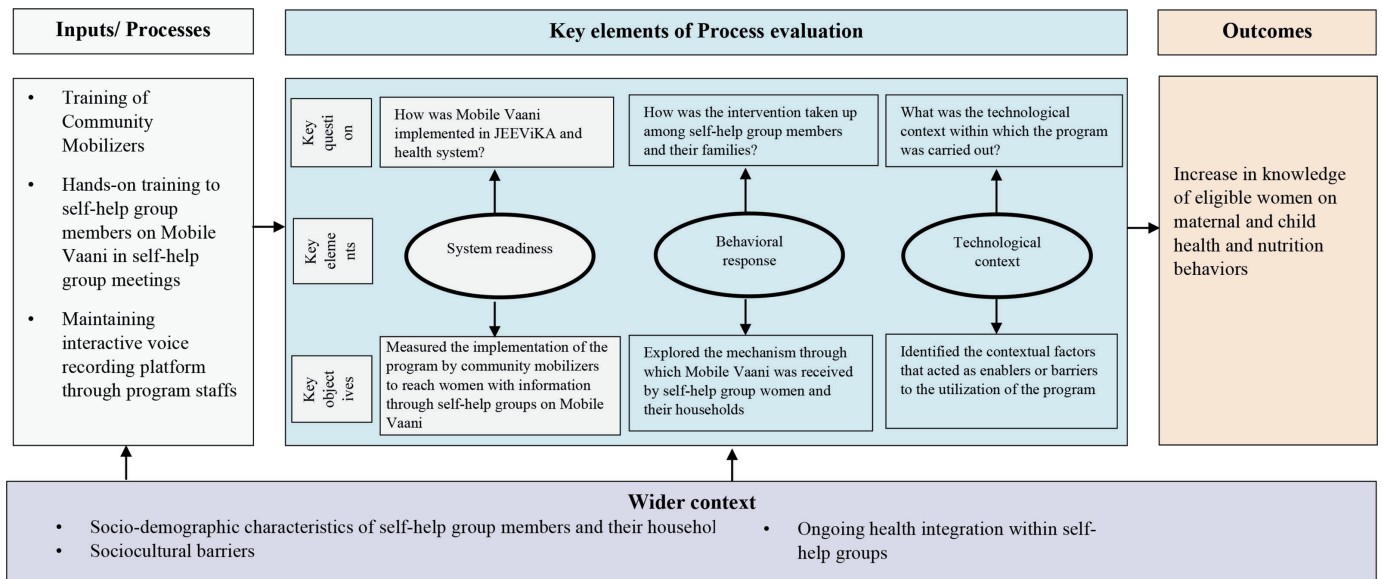

**Figure 1** Conceptual framework outlining key components of process and outcome evaluation.

understand Mobile Vaani and share its key tenets with their members, (2) a behavioural response to the programme, as determined by the self-help group members' receptivity to accept, use and disseminate Mobile Vaani with others; and (3) the technological context within which the intervention was being implemented, that acted as an enabler or barrier to the uptake and utilisation of the programme (figure 1).[66 67]

### Data collection and sampling

#### Outcome evaluation: cross-sectional surveys

For the outcome evaluation, cross-sectional surveys were carried out at two time points: (1) baseline survey (n=2436) from October 2017 to January 2018; and (2) endline survey (n=2498) 12 months following implementation from November 2018 to January 2019 (table 1). Cross-sectional surveys were powered to detect at least 7% net change in knowledge on core themes among the target population after 12 months of intervention. Assuming 80% power, and a design effect of 1.5, a sample size of 1200 self-help group households in each arm was sufficient to detect this change. Study respondents were eligible women, aged 15–49 years, currently married, having had a live birth in the last 24 months preceding the survey and were from self-help group households. At endline, women were cross sectionally selected from the same self-help groups. Less than 1% of the data collected across the variables of interest were missing and they were missing completely at random.

In the surveys, the women were asked about their engagement with Mobile Vaani and the intervention's impact on their knowledge of: maternal and child nutrition, family planning and management of diarrhoea. Data were collected by female enumerators (n=40) with prior experience in conducting similar surveys. Enumerators were grouped in teams of five, including one designated mentor per team, and a supervisor who accompanied

them to households, supported logistics, and checked completed study questionnaires. All data were captured on mini-laptops using CS-Pro V.7.1 software. Quality control occurred at three levels: (1) supervisory oversight of individual data collectors; (2) 10% spot checks of interviews by a team mentor; and (3) outlier analyses of entered data by the core research team.

*The process evaluation* was achieved through a triangulation of three primary data sources that were collected between November 2017 and November 2018 and were used to measure the system's readiness and the behavioural response to the intervention (table 1). Secondary data were analysed to better understand the technological context around the intervention.

#### Process evaluation: system readiness

System readiness was measured through surveys with community mobilisers and observing self-help group meetings where community mobilisers talked about Mobile Vaani with their members. Surveys with 116 community mobilisers were carried out among those community mobilisers who had attended at least two trainings with field officers. After obtaining written consent from the respondents, data were captured in Hindi using computer-assisted personal interview software on laptops by trained female enumerators. Direct observations of self-help group meetings were conducted with the same community mobilisers who had been interviewed previously. These observations helped document what and how community mobilisers shared information about Mobile Vaani with their members. Trained investigators filled in structured questionnaires. Verbal consent was sought from all participants.

#### Process evaluation: behavioural response

Behavioural response was captured through 180 qualitative interviews carried out among selected self-help group

**Table 1** Details of data collection points

| Data source | Study population | Sample | Date of data collection | Key outcomes | Key explanatory variables | Control variables | Presentation of results |
|---|---|---|---|---|---|---|---|
| **Outcome evaluation** | | | | | | | |
| Structured quantitative interviews | Eligible women | Baseline: 2436 women | Baseline: October 2016–January 2017 Endline: November 2018–January 2019 | Knowledge on: Complementary feeding (Yes/No), Correct consistency of food for a child 6–12 months old (Yes/No), How to make a child's food nutrient rich and dense (Yes/No), Elements of adequate nutrition for a pregnant or lactating woman (Yes/No), Two modern spacing family planning methods (Yes/No), FLW as a source of short-term FP methods (Yes/No), How to identify and manage diarrhoea among children (Yes/No) | Study arm (Intervention and comparison arm) Survey time (baseline and endline) | Women's parity (average), Number of completed years of formal education (no formal education, 1–5 years, 6–9 years, ≥10 years), Any exposure to mass media (ie, radio, television or newspaper) over the past year (Yes/No), Religion (Hindu, Others) Scheduled caste/tribe (Yes/No), Household wealth (five quintiles), Self-help group membership (Yes/No), Duration of membership (range: 0–180months), Living in a nuclear family (Yes/No), Contact with a frontline worker during pregnancy (Yes/No), Contact with a frontline worker within 7days of delivery (Yes/No) | Tables 2 and 3 |
| **Process evaluation** | | | | | | | |
| Structured quantitative interviews | Community mobilisers | 116 | November 2017–November 2018 | Knowledge score of community mobilisers around Mobile Vaani themes (range: 0–31) | Number of trainings received by community mobilisers on health and nutrition themes (0–1 theme, 2 themes, 3–4 themes) | Age (mean age) Caste (SC/ST, others), Education (1–10th std, 11th–12th std, college/more), Engaged in other occupation (yes/no), Duration of working as community mobiliser (0–1 year, 3–4 years, 5 or more years), Exposure to mobile phone (no/partial exposure, full exposure, that is, having a personal phone, having access to a phone at all times, and ability to dial a number) | Table 4 |
| Direct observations | Self-help group meetings | 116 | November 2017–November 2018 | Community mobiliser's ability to dial a number (Yes/No), Community mobiliser's ability to explain Mobile Vaani features to women in a self-help group meeting (Yes/No) | | | |
| In-depth qualitative interviews | Community mobilisers self-help group members Programme staff | 55 180 18 | November 2017–November 2018 | Integrating Mobile Vaani into self-help groups Content preferred by users Type of content recollected and shared by respondents Reported barriers to accessing Mobile Vaani | | | Table 5 |
| System generated interactive voice recording data | Phone numbers | 2.6 million calls 44664 unique phone numbers/users | February 2017–July 2018 | Missed calls—user called the system but the system did not call back, Call backs—system called the user following a missed call from the user, Outbound calls—system-generated random calls to registered numbers at periodic intervals | | Answered calls—user answered incoming call, Unanswered calls—user did not answer incoming call, Duration of listenership, in minutes, Reasons for disconnecting the call—network issues, Mobile Vaani issues, user issues | Figures 2 and 3 |

FLW, frontline worker; FP, family planning.

women who knew of Mobile Vaani. They were interviewed in Hindi by trained investigators. Data were captured through audio recordings after obtaining written consent from the respondents. The individual interviews lasted an average of 40 min. Audio files were transcribed and then translated to English. Transcripts were imported into a qualitative software programme, Atlas.ti V.6.2, for coding and in-depth analysis.

### Process evaluation: technological context

The technological context was captured through secondary analyses of system-generated interactive voice recording data, which included 2.6 million call records from 44 664 unique phone numbers. Call records, take took place from February 2017 to July 2018, were obtained through the implementors and were analysed to study calling and listening patterns and to identify any potential technical hindrances or barriers users may have faced.

### Variables (Table 1)

For the outcome evaluation, seven key dependent variables were measured. These included women's knowledge on three indicators related to child nutrition, namely: that complementary feeding for an infant was to be initiated between 6 and 12 months, what the correct consistency of food for a child 6–12 months old was, and how a child's food could be made nutrient and energy dense. The indicator capturing knowledge around maternal nutrition included the elements of adequate nutrition for a pregnant or lactating woman. Two indicators on family planning captured the respondents' knowledge of at least two modern spacing methods of family planning, and knowledge that frontline workers could provide short-term family planning methods, for example, condoms/pills to women in need. The seventh indicator captured the interviewees' knowledge of the symptoms of diarrhea and its management. There were two key dichotomous independent variables of interest; one was the study arm, that is, intervention or comparison, and the other was the survey round, that is, baseline and endline. Control variables were included as these indicators were known to influence findings around the key dependent variables in previous studies.[1 68 69] These indicators were: women's parity (captured as an average measure across all the women in the category), number of completed years of formal education (categories: no formal education, completed 1–5 years of schooling, 6–9 years of schooling and 10 or more years), any exposure to mass media over the past year (yes/no), belonging to the Hindu religion/not, belonging to a scheduled caste/tribe or not, household wealth (measured in quintiles) (The household wealth index was developed through principal component analysis using information on 26 household assets measured for six categories, five housing characteristics and one on asset ownership, taken from a nationally representative tool. Using these data, wealth quintiles were developed based on equal proportions of the population into five categories, from the first quintile being the poorest to the fifth quintile being the richest.), respondent is a self-help group member/not, average duration of membership (in months), living in a nuclear family structure/not and if the respondent had any contact with a frontline worker during pregnancy and 7 days postpartum.[1 68 69]

### Process evaluation: system readiness

In order to determine whether the community mobilisers were sharing adequate information about Mobile Vaani in their groups, community mobilisers were assessed on their knowledge around four key topics Mobile Vaani intended to share with self-help group women. This was captured through 31 knowledge-related questions asked to each of the community mobiliser with a correct answer to each question garnering a point; the scores for each of the responses were summed up to form a continuous variable. Through observations of self-help group meetings, community mobilisers were also assessed on whether they could demonstrate to their self-help group members how to dial a number on a mobile phone, how to access Mobile Vaani and the key features of the programme. Other variables of interest were the number of messages community mobilisers reported being trained on (maximum being four: child nutrition, maternal nutrition, family planning and management of diarrhoea); and other characteristics of the community mobilisers, that is, age (captured as an average), caste (belonging to a scheduled caste/tribe or not), education (completed 1–10 years of schooling, 11–12 years of schooling and college/more), whether she is engaged in another occupation or not, number of years she has been a community mobilisers (0–2 years, 3–4 years, 5+ years), and exposure to none/partial access to a mobile phone or complete access to a mobile phone. (Full exposure to mobile phone meant owning a phone or having access to one 24 hours a day, and knowing how to dial a number. Partial access meant one of the elements of full exposure were missing.)

### Process evaluation: behavioural response

From the qualitative interviews that were held with women, four broad themes were identified and are presented in this article as they shed light on the outcome evaluation and help explain some of the enablers and barriers to the uptake of the programme. These themes include the women's perspectives on how Mobile Vaani was integrated into self-help group meetings, the content they preferred listening to, the stories they resonated with and shared with others, and some of the barriers they faced in accessing the programme. Key indicators such as the age and educational level of the women who were interviewed were also captured.

### Process evaluation: technological context

The purpose for analysing the system-generated interactive voice recording data was to capture the lifecycle of a phone number within the system. The key variables included: a missed call, that is, the user called the

programme generating a missed call, but the call was never answered back; a call back where a missed call from the user generated a call back from the system within 30 min of the user calling the system; and, an outbound dial, where the system called a registered user at periodic intervals on its own accord. In addition, the calls sent by the system were further analysed to determine whether they were answered or not answered by the users. If the calls were answered, the duration for which the users listened to the content were categorised in seconds and minutes. Further, reasons for why calls were not answered were categorised into network issues, problems generating from the programme itself or from the user, such as, the user not picking up the call.

## Statistical procedures
### Outcome evaluation: cross-sectional surveys
Differences between women's individual characteristics, that is, covariates, across the intervention arms between baseline and endline were tested using ordinary least-squares (for continuous variables) and through logistic regression models (for categorical variables) (table 2). For instance, women's parity and self-help group membership duration were measured as continuous variables, and hence tested using ordinary least-squares regression. The remaining variables were categorical, and hence tested using logistic regression model. In addition, the findings were adjusted for random clustering effect at the block and village levels by using multilevel mixed-effect regression analyses.

To assess the net change in the seven outcome variables of interest from baseline to endline, two types of multivariate analyses were carried out: one, a difference-in-difference analysis and two, a treatment-on-the-treated approach was applied using instrumental variable analysis (table 3). The instrument variable was those who were exposed to Mobile Vaani—as captured by those women who recalled hearing about Mobile Vaani or who recalled listening to content on it—thus controlling for endogeneity to programme exposure.[70–72] These analyses were adjusted for covariates and for random clustering effect at the block and village levels. Samples with missing data were dropped from the multivariate analyses as the sample lost was negligible. All analyses were done using Stata V.16.0 software programme.

### Process evaluation: system readiness
Bivariate and descriptive analyses were conducted on key indicators from the quantitative interviews conducted with community mobilisers and observations from self-help group meetings (table 4). The aim of the analyses was to capture the association of three key indicators that assessed the capabilities of the community mobilisers (knowledge on Mobile Vaani themes, ability to dial a number and explain all the features of the programme) with other characteristics of interest, for example, their level of training, their sociodemographic profile, etc.

### Process evaluation: behavioural response
The interview guides that the investigators used in the field covered several topics that were of interest to the evaluation. Once the interviews had been transcribed and imported into the qualitative software programme, the transcripts were tagged with 70 codes that were generated from coding the transcripts. The codes were then brought together under nine broad themes of which four relevant themes are shared here.

### Process evaluation: technological context
To analyse system-generated data, large volumes of data on phone calls to and from the system were merged. Cascade analyses of unique mobile numbers were conducted to determine user's engagement with the Mobile Vaani platform over time. For this, a lifecycle of each phone number was created. While crafting out the journey of each unique number's lifecycle, all the calls made by or to Mobile Vaani by the number were considered as exposures to Mobile Vaani. Further, durations of each call and reasons for a call not going through were also studied.

## RESULTS
### Outcome evaluation
Table 2 present summary characteristics for the sample population used in the cross-sectional survey. Findings suggest that more than half the women had no formal education, >90% belonged to Hindu religion, and >30% were members of the most disadvantaged schedule caste/tribe. While the majority of the sample population reported having had contact with a frontline worker (FLW) during their last pregnancy, less than half had a postnatal visit with a FLW 7 days post delivery. Self-help group membership was generally high with about 61%–78% of the respondents in each of the study arms being a member themselves; the remaining eligible women interviewed came from a household where another adult female family member was a self-help group member. On average, self-help group members had been associated with their group for almost 3 years. For most sociodemographic characteristics assessed, the differences between study arms were not statistically significant suggesting that randomisation was successful for the study design.

At endline, 17% (n=223) of the women in the intervention arm could recall listening to Mobile Vaani content at least once. Among them, most recalled listening to content on the themes of maternal nutrition (38%), child nutrition (35%), family planning (13%) and diarrhoea management (12%). Health and nutrition messages were reported by 77% of women listeners to be the most useful.

Table 3 depicts results from the difference-in-difference and treatment-on-treated effect of the Mobile Vaani intervention on knowledge for selected maternal and child health practices. Women's knowledge of the correct consistency of food for a child 6–12 months of age was significantly higher over time across study arms (difference-in-difference: 6.6% (1.1%–12.2%) p=0.019)

**Table 2** Sociodemographic characteristics of the women interviewed in the cross-sectional surveys of the outcome evaluation

| Sociodemographic characteristics | Baseline | | | Endline | | |
|---|---|---|---|---|---|---|
| | Intervention (A) (N=1249) | Comparison (B) (N=1187) | P value A versus B | Intervention (C) (N=1188) | Comparison (D) (N=1310) | P value A versus C |
| Women's parity, | | | | | | |
| Mean (SD) | 2.95 (1.68) | 3.06 (1.74) | 0.142 | 2.81 (1.58) | 2.91 (1.65) | 0.008 |
| Women's education, % | | | | | | |
| No formal education | 56.0 | 57.5 | 0.535 | 47.0 | 49.4 | <0.001 |
| 1–5 years | 12.3 | 14.5 | 0.112 | 13.9 | 14.7 | 0.200 |
| 6–9 years | 15.8 | 15.2 | 0.729 | 20.0 | 20.2 | 0.005 |
| ≥10 years of schooling | 15.9 | 12.8 | 0.064 | 19.2 | 16.7 | 0.020 |
| Any mass media exposure, % | | | | | | |
| No | 62.9 | 74.5 | <0.001 | 53.4 | 63.0 | <0.001 |
| Yes | 37.1 | 25.5 | <0.001 | 46.6 | 37.0 | <0.001 |
| Belongs to Hindu religion, % | 98.5 | 91.1 | <0.001 | 99.2 | 90.8 | 0.136 |
| Households' wealth quintile, % | | | | | | |
| First | 13.6 | 28.4 | <0.001 | 15.4 | 24.7 | 0.259 |
| Second | 18.3 | 21.1 | 0.126 | 19.6 | 20.0 | 0.383 |
| Third | 20.3 | 19.0 | 0.468 | 20.5 | 19.4 | 0.870 |
| Fourth | 23.8 | 15.8 | <0.001 | 23.0 | 17.3 | 0.669 |
| Fifth | 24.1 | 15.7 | <0.001 | 21.5 | 18.6 | 0.122 |
| Scheduled caste/tribe, % | | | | | | |
| No | 68.6 | 69.3 | 0.815 | 69.6 | 65.5 | 0.654 |
| Yes | 31.4 | 30.7 | 0.815 | 30.4 | 34.5 | 0.654 |
| Woman herself is a self-help group member, % | | | | | | |
| No | 31.9 | 21.9 | <0.001 | 39.3 | 33.5 | <0.001 |
| Yes | 68.1 | 78.1 | <0.001 | 60.7 | 66.5 | <0.001 |
| Women's self-help group membership duration in months | | | | | | |
| Mean (SD) | 34.8 (20.1) | 26.9 (17.9) | <0.001 | 38.5 (22.6) | 36.1 (20.5) | 0.001 |
| Nuclear family structure, % | | | | | | |
| No | 67.4 | 57.3 | <0.001 | 71.6 | 65.0 | 0.007 |
| Yes | 32.6 | 42.7 | <0.001 | 28.4 | 35.0 | 0.007 |
| Any contact with FLW during pregnancy, % | | | | | | |
| No | 27.8 | 43.0 | <0.001 | 35.7 | 37.9 | <0.001 |
| Yes | 72.2 | 57.0 | <0.001 | 64.3 | 62.1 | <0.001 |
| Any contact with FLW within 7 days of delivery, % | | | | | | |
| No | 58.1 | 54.8 | 0.243 | 55.8 | 48.2 | 0.282 |
| Yes | 41.9 | 45.2 | 0.243 | 44.2 | 51.8 | 0.282 |

Differences in groups at baseline and endline were tested by using ordinary least-squares regression models (continuous variables) or logit regression models (categorical variables), adjusting for clustering effect at the block and village level.

but not based on awareness of Mobile Vaani (treatment-on-treated: 14.4% (−6.1%–35%), p=0.169). Women's awareness of the need to make a child's food nutrient and energy dense increased significantly across study arms (difference-in-difference: 9.3% (4.5%–14.1%) p≤0.001) and among those with any awareness of Mobile Vaani (treatment-on-treated: 18.8% (0.4%–37.2%) p=0.045).

Women's knowledge of proper nutrition for pregnant or lactating women increased in both groups over time but more substantially in the comparison arm from baseline to endline (intervention: 73.6%–77.0%; control: 69.5%–80.9%; difference-in-difference: −8.0% (−12.7% to −3.3%) p=0.001). Significant differences were not observed among those with any awareness of Mobile Vaani

**Table 3** Results from treatment-on-the-treated effect of Mobile Vaani intervention on knowledge of selected maternal and newborn health practices for outcome evaluation, 2017–2019

| Indicators | Intervention | | Comparison | | Difference-in-difference estimate | P value | Treatment-on-treated estimate | P value |
|---|---|---|---|---|---|---|---|---|
| | Baseline (N=1249) | Endline (N=1188) | Baseline (N=1187) | Endline (N=1310) | | | | |
| **Knowledge indicators, %** | | | | | | | | |
| Need to initiate complementary feeding for an infant at 6–12 months | 82.5 | 79.6 | 83.5 | 84.4 | −3.8 (−8.0, 0.4) | 0.077 | −12.1 (−25.2, 0.9) | 0.069 |
| Correct consistency of food is for a child aged 6–12 months | 55.6 | 56.4 | 56.0 | 50.2 | 6.6 (1.1, 12.2) | 0.019 | 14.4 (−6.1, 35) | 0.169 |
| How to make a child's food nutrient and energy dense | 72.6 | 79.8 | 74.3 | 72.2 | 9.3 (4.5, 14.1) | <0.001 | 18.8 (0.4, 37.2) | 0.045 |
| Elements of adequate nutrition for a pregnant or lactating woman | 73.6 | 77.0 | 69.5 | 81.0 | −8.0 (−12.7, −3.3) | 0.001 | −0.3 (−16.8, 16.2) | 0.971 |
| Can name at least two modern spacing methods of family planning | 90.3 | 89.4 | 84.8 | 88.6 | −4.7 (−8.2 to −1.2) | 0.009 | 17.6 (4.7, 30.5) | 0.008 |
| Aware that frontline worker carries condoms and pills | 8.1 | 14.9 | 9.2 | 12.8 | 3.2 (−0.3, 6.7) | 0.077 | 2.8 (−9.3, 15) | 0.647 |
| Knows symptoms of diarrhoea and how to manage it | 12.1 | 17.9 | 16.5 | 16.2 | 6.0 (2.0, 9.9) | 0.003 | −15.2 (−30.9, 0.4) | 0.056 |

Estimates are adjusted for women's parity, education, religion, caste, family structure, wealth quintile, mass media exposure, respondent's membership in groups and duration of self-help group membership, FLW contacts during pregnancy and after delivery, accounting for clustering effect at block and village level.
Treatment-on-treated effect is estimated based on instrument variable regression analysis and adjusted for women's parity, education, religion, caste, family structure, wealth quintile, mass media exposure, respondent's membership in groups and duration of self-help group membership, FLW contacts during pregnancy and after delivery, accounting for clustering effect at block and village level.

as compared with those with no awareness (treatment-on-treated: −0.3% (−16.8%–16.2%) p=0.971). Women's awareness of at least two modern spacing methods of family planning was significantly higher in the comparison arm across study arms (difference-in-difference: −4.7 (−8.2%–1.2%), p=0.0009) but when exposure to Mobile Vaani was considered, the opposite trend was observed and, a 17.6% (4.7%–30.5%; p=0.008) increase in knowledge was observed among those with any awareness of Mobile Vaani. Any awareness of the symptoms of diarrhoea in children under 2 years of age and its management was low overall but increased significantly across study arms (difference-in-difference: 6.0% (2.0%–9.9%) p=0.003). However, these findings did not stay significant when knowledge of symptoms and management of diarrhoea was considered among those with awareness of Mobile Vaani.

### Process evaluation: system readiness

Table 4 depicts the characteristics of community mobilisers who were responsible for leading self-help groups and sharing information with self-help group members on Mobile Vaani. Community mobilisers were an average of 34 years of age, with 56% having more than 10 years of formal schooling, and 13% coming from the disadvantaged scheduled caste/tribe groups. Community mobilisers had similar knowledge scores irrespective of background characteristics. However, those who had received two or more trainings could recall more messages correctly. Community mobilisers who had worked for a greater duration of time (three/more years) in their role and those with higher education (more than 10 years of schooling) had greater competency in dialling a number and explaining Mobile Vaani to their members than their counterparts.

### Process evaluation: behavioural response

Findings from in-depth interviews on the integration of Mobile Vaani into self-help groups provided insights into the barriers and facilitators to Mobile Vaani use (table 5). Community mobilisers were hailed as the key peer educators to integrate Mobile Vaani into self-help groups, by sharing various aspects of the intervention with members in self-help group meetings (theme 1). There was a general affinity towards the content shared through the intervention and reasons cited for this included it being their only source of news, and a useful medium for information on maternal health, nutrition and sanitation, which they could access at any time free of cost (theme 2).

When it came to preferences, while most women expressed that they enjoyed listening to stories of local women from neighbouring communities, their perceptions on the prerecorded stories and messages generated in the studio were mixed with some younger and more educated women enjoying the messages that were woven into hypothetical scenarios with older and less educated women finding the content, language and dialect hard to understand and the pace of the storyline too fast, in

**Table 4** Findings showing association of characteristics of community mobilisers to sharing Mobile Vaani in self-help group meetings (N=116)

| Community mobilisers' background characteristics | Ability of community mobilisers to give training on proposed topics during self-help group meetings | | | |
|---|---|---|---|---|
| | Community mobilisers observed giving Mobile Vaani training in self-help group meetings | Average knowledge score range, as captured during an interview (0–31) | Dial a number, as noted in a training/ self-help group meeting | Explain Mobile Vaani features to self-help group members in a meeting |
| | N (%) | Mean (SD) | Proportion (%) | Proportion (%) |
| Number of health and nutrition messages that community mobilisers received training on | | | | |
| 0–1 theme | 14 (12.1) | 8.2 (1.11) | 57.1 | 64.3 |
| 2 themes | 80 (69.0) | 8.6 (1.22) | 41.3 | 63.8 |
| 3–4 themes | 22 (19.0) | 8.9 (0.87) | 31.8 | 45.5 |
| Age (mean, SD) | 33.3 (8.45) | 8.6 (1.16) | 31.7 (8.18) | 33.0 (7.95) |
| Caste, % | | | | |
| SC/ST | 15 (12.9) | 8.5 (1.21) | 46.7 | 66.7 |
| Others | 101 (87.1) | 8.6 (1.15) | 40.6 | 59.4 |
| Education, % | | | | |
| 1st–10th standard | 51 (44.0) | 8.5 (1.05) | 29.4 | 54.9 |
| 11th–12th standard | 39 (33.6) | 8.7 (1.29) | 46.2 | 64.1 |
| College and above | 26 (22.4) | 8.7 (1.15) | 57.7 | 65.4 |
| Engaged in other occupation, % | | | | |
| Yes | 36 (31.0) | 8.5 (1.03) | 50 | 69.4 |
| No | 80 (69.0) | 8.6 (1.19) | 37.5 | 56.3 |
| Duration of work as a community mobiliser (years), % | | | | |
| 0–2 years | 32 (27.6) | 8.5 (1.23) | 28.1 | 31.3 |
| 3–4 years | 48 (41.4) | 8.8 (0.96) | 43.8 | 68.8 |
| 5+ years | 36 (31.0) | 8.5 (1.27) | 50 | 75 |
| Exposure to mobile phone, % | | | | |
| No/partial exposure | 8 (6.9) | 8.7 (1.55) | 37.5 | 50 |
| Full exposure | 108 (93.1) | 8.6 (1.11) | 41.7 | 61.1 |

SC/ST, schedule caste/tribe.

general (theme 3). Despite this, most women who were interviewed shared either the Mobile Vaani number or the information heard on Mobile Vaani at least once with their family members, neighbours and relatives with a few recounting stories of using the information they had learnt to help others in dire need.

Qualitative findings suggest that there were economic, gendered and comprehension-related barriers, which hampered the uptake of the intervention (theme 4). The most common reasons women shared that they did not listen to Mobile Vaani were a lack of time and no apparent benefit. Some women said that they had insufficient balance in their phones to give a missed call and a few women expressed doubts about Mobile Vaani being a free service. Beneficiaries also noted that calls came without warning at timings that were not always convenient or when women had access to mobile phones. Some respondents with migrant husbands found it problematic to be on a Mobile Vaani call as their spouses complained that their (women's) phones were busy when the husbands called. Women also expressed apprehension in using mobile phones for long durations in front of male members of the household.

**Technological context: barriers to use**

Beneficiaries could access Mobile Vaani either by answering an outbound call sent by Mobile Vaani or by directly giving a missed call (inbound call) and then being called back by the system. A review of the call records of the 44664 users suggests that over a few exposures, the proportion of users receiving call-backs decreased, while the proportion of users who received random outbound calls being generated by the system increased (figure 2).

An analysis of 2.6 million call records showed that around 19% of the calls were missed calls generated by the beneficiaries and 81% of all calls were made by Mobile

**Table 5** Quotes from in-depth interviews with 180 self-help group women

| Thematic reference in paper | Characteristics of women quoted | Quote |
|---|---|---|
| Theme 1: integrating Mobile Vaani into self-help groups | 30 years old, 12 years of education | *Community mobilizer is a homemaker. So, she spoke to us in the local dialect. We were all able to understand how we are supposed to dial the number, including the illiterate women of the(self-help) group.* |
| Theme 2: content preferred by users | 38 years old, 10 years of education | *I mostly like the voice of different women from villages as I feel motivated by listening to their stories. I also feel I should record my own voice just like they do.* |
| | 35 years old, 9 years of education | *I like the studio voice more. The village women speak in breaks and they don't speak very well. But the studio sister speaks well and explains things very nicely.* |
| Theme 3: type of content recollected and shared by respondents | 30 years old, 3 years of education | *I have also discussed [about what I hear on Mobile Vaani] with my mother-in-law and sisters-in-law. My younger sister-in-law has a small baby and she needs to feed her baby twice or thrice in a day. She should feed her baby properly so the baby will remain healthy.* |
| Theme 4: reported barriers to accessing Mobile Vaani | 35 years old, 8 years of education | *You know how husbands are! They will start a fight with us if we are always busy with the mobile. I use it only in his absence and don't even touch it when he is home.* |

Note: these quotes are an illustrative sample from a universe of 180 in-depth interviews conducted with self-help group women who had heard content on Mobile Vaani and engaged with the programme in meetings as well.

Vaani to the users (figure 3). Of these 81%, the majority were outbound calls (62%) made randomly by the system while the remaining 38% were call-backs generated as a result of a beneficiary sending a missed call to the system. Users' response to outbound dialled calls and call-backs were explored further and found that even though the system made more outbound calls than call-backs, the proportion of users answering the calls was much higher for call-backs as compared with outbound dialled calls, suggesting that call-backs were more cost effective for Mobile Vaani. If calls were initiated by users, 50%–60% of the calls were answered as compared with calls that were pushed by the system (20%–25%). A quarter of outbound dialled calls were disconnected within 30 s of answering, and users spent longer listening to messages on call-backs. Majority of the calls, irrespective of how they were received, were disconnected because of issues at the users' end which included 'normal clearing/no answer/call rejected/user busy'. Additionally, the reasons

for call disconnection at Mobile Vaani end were network congestion, incapacity to handle a large volume of calls by primary rate interface lines, and other technical problems within the Mobile Vaani system (figure 3).

## DISCUSSION

This study presents results from a mixed methods evaluation of the Mobile Vaani programme in two districts of Bihar. From the outcome evaluation, 23% of women from households with a self-help group member reported having heard about Mobile Vaani and 17% reported listening to content within the 12 months preceding the endline survey. Among women exposed to Mobile Vaani, qualitative findings suggest that they reportedly liked the programme, found it convenient to access, and enjoyed recording and sharing content with friends and neighbours. However, the programme was met with several existing contextual barriers such as lack of mobile

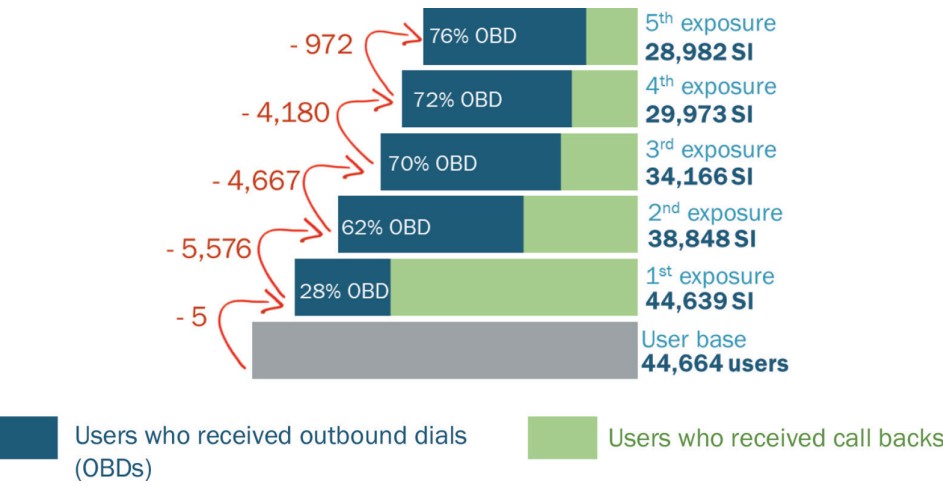

**Figure 2** Cascade of unique numbers engaging with Mobile Vaani over the course of the programme.

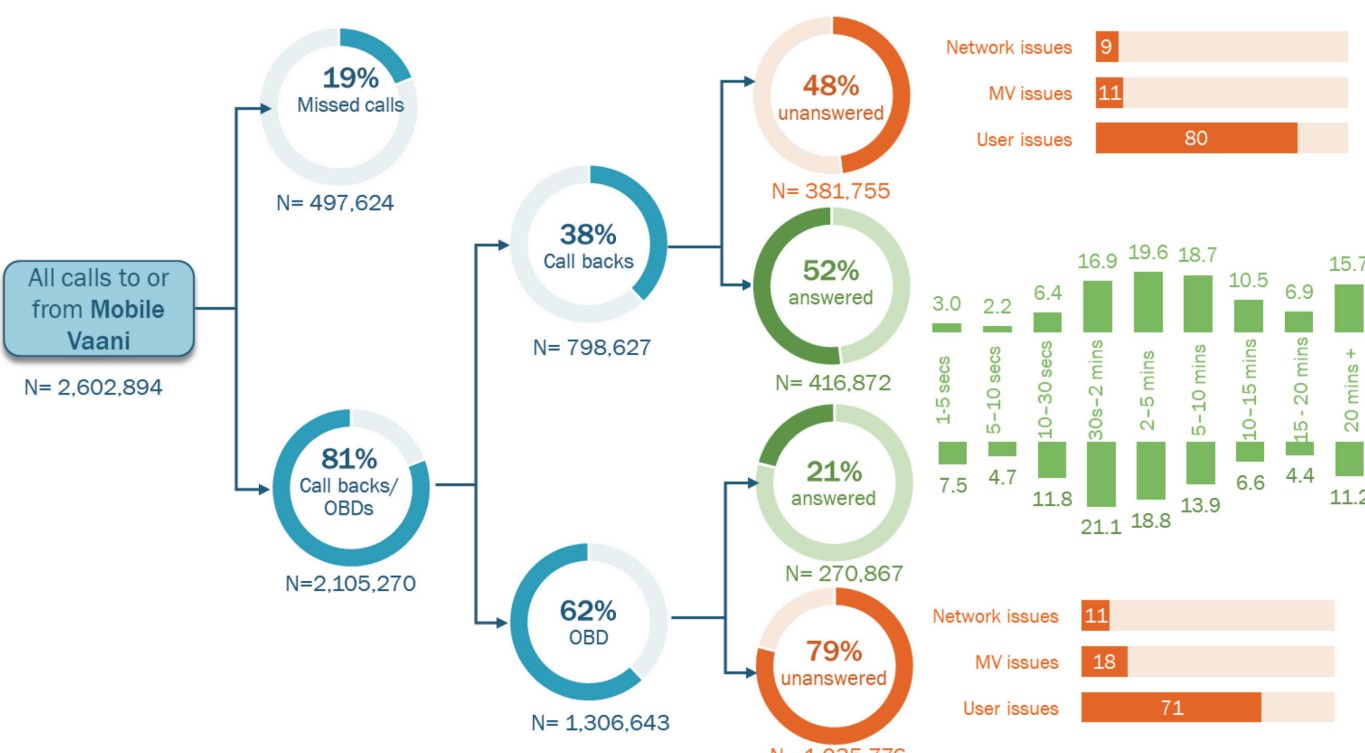

**Figure 3** User's engagement with Mobile Vaani system.

literacy, poor phone infrastructure, inability to use mobile phones, gender and cultural barriers, lack of time, among others.[73] The programme's impact on women's knowledge was mixed and limited to child nutrition and modern contraceptive methods. Overall findings underscore the challenges of trying to reach poor and marginalised groups of women with limited access to and skills to use mobile phones with digital interventions.

Study findings contribute to an emerging body of evidence on direct-to-consumer mobile health interventions in India.[74 75] The mobile messaging service Kilkari which aims to provide new and expectant mothers and their families with health information messages across 13 states in India similarly had a significant impact on a few but not all health behaviours assessed, namely reversible modern contraceptive methods and child immunisations at 10 weeks.[76] The limited impact observed there and in our evaluation of Mobile Vaani suggests that a more targeted approach to content delivery and one which considers differentials in digital access and skills may deepen impact.

Changing behaviours within communities takes time, concerted effort and reiteration of information on correct health practices, which can be achieved through repeated messaging and voice reminders using mobile phone technology through self-help groups.[19 77 78] However, it is important to note that health interventions using mobile phone technology may not function effectively in isolation but instead could be incorporated as a tool for improving indicators within a holistic health systems approach. In addition, community mobilisers also serve as additional conduits to reaching women with key health

and nutrition messaging, besides frontline workers who often bear the burden of being the primary interlocutors on health issues.[79–81] However, care should be taken not to overburden the community mobilisers whose primary role is to support self-help groups.

Mobile Vaani's impact on knowledge was driven by programme coverage and exposure which were assessed based on reported awareness of and engagement with the programme. Future evaluations might consider linking system generated data, including call data records, with survey data assessing outcomes. Population level coverage of Mobile Vaani among pregnant women was 23% based on reported survey estimates. An effort to validate this measure using call data records yielded a similar estimate. While the Kilkari programme enrolled targeted women using government tracking registries, their overall population level coverage was 21% highlighting the challenge in reaching key audiences through mobile phone numbers within the Indian context.[82 83]

Reported exposure to Mobile Vaani was based on self-reported estimates and not call data records because of challenges linking varied data sources. Exposure was impeded by the limited number of self-help group members with mobile phones and children under 2 years of age (target population). Broader systematic challenges in women's access to and use of mobile phones persist in Bihar and elsewhere throughout India.[80 84 85] Women in the study sample reported having limited access to mobile phones, restricted by work and domestic commitments, coupled with limited digital skills. Further, patriarchal gender norms were omnipresent and restricted their use of mobile phones.[85 86] Similar contextual factors

have been highlighted in explaining low outreach of mass media as well, including newspapers, radio and television, among women in rural Bihar.[1 2 26]

The evaluation highlighted the possibility to scale up the intervention with several caveats in mind. *First,* a space would need to be identified where a wide community could be reached that promoted ownership and sustenance of the intended intervention. *Second,* evidence-based and systematic planning would be needed to design a robust programme to achieve its desired take-up by reaching intended audiences through capacity building and extensive promotion of the programme. Adequate coverage of the programme would be essential for success. *Third,* the content being played through the interactive voice recording system would need to focus on topics of interest and be relatable to the audience to sustain its programming. As the audience would expand to new populations, this would require additional formative research and continued monitoring to tweak the programme and allow for mid-course corrections. *Fourth,* for optimal results, the programme ought to coordinate its content development and dissemination with any parallel programme efforts disseminating health and nutrition information to the intended audience. *Fifth,* the programme would need to identify and account for various sociodemographic and economic differences with the intent of reaching different audiences equally and not exacerbating disparities of access across socioeconomic characteristics. Other scholars have also captured this phenomenon of intersecting inequalities, especially in terms of gendered phone access gaps and financial constraints.[46 73 80 87] *Sixth,* a robust registration and follow-up system of registered phone numbers would be pivotal in ensuring that messages would reach their intended audience at the appropriate time.

Though intervention and comparison arms were matched successfully, with the treatment and control groups generally comparable for their basic characteristics, this evaluation did have some limitations. Baseline and endline samples were not always balanced across the intervention and comparison arms, but key covariates were controlled for and accounted in the analysis. The study used fresh cross-sections of respondents at both time points of survey; however, respondents at endline belonged to the same self-help groups that were included at baseline, and therefore were representative of similar geographical and socioeconomic conditions. As the study only engaged self-help group members, it may not have considerable external validity outside of this population. The exposure among the intended audience was low, but this was adjusted for in the analyses using an instrumental variable of recalled exposure to the programme. The recall bias among respondents was also accounted for through rigorously trained investigators and pilot testing the survey tools, which were formulated based on established modules from similar surveys.

## Conclusion

When introduced to Mobile Vaani through self-help groups, women's knowledge of key health and nutrition outcomes improved for two out of seven key indicators. The study provides evidence that mobile-based interventions could evolve into a modality to reach marginalised populations through systematic planning when coupled with other interventions on the ground. It would further be advisable for programmes to remain conscious of the context, and enablers or barriers driven by socioeconomic and gendered realities of the target audience, before operationalising capital intensive interventions.

### Author affiliations
[1]Population Council, New York, New York, USA
[2]Population Council, Delhi, India
[3]Johns Hopkins University Bloomberg School of Public Health, Baltimore, Maryland, USA
[4]Bill & Melinda Gates Foundation, Seattle, Washington, USA
[5]Gram Vaani, Delhi, India
[6]Project Concern International, San Diego, California, USA
[7]Department of Rural Development, Government of Bihar, JEEViKA, Bihar Rural Livelihoods Promotion Society, Patna, India
[8]University of Cape Town, School of Public Health and Family Medicine, Cape Town, Western Cape, South Africa

**Contributors** LI served as the PI of the study designing the evaluation, overseeing the data collection, leading the analysis and interpretation of results. LI was the primary author of the paper, had full access to the data, controlled the decision to publish and accepts full responsibility for the conduct of this study, as the guarantor. S, RM, RKV, AN and SS led the data collection, contributed to data analysis and visualisation, and wrote early drafts of the methods and results sections of the manuscript. DM contributed to data analysis, interpretation of results and critical revision of the manuscript. DD and AELF contributed to the plan and design of the study, contributed to data analysis, interpretation of results and critical revision of the manuscript for important intellectual content. AS, IC, MRC, AP, AG contributed to designing the study, assisted the evaluators to collect primary data and access secondary data, and provided critical insight into the interpretation of the results and revisions of the manuscript. All authors have reviewed and approved the final version of the manuscript.

**Funding** This study was supported by the Bill and Melinda Gates Foundation [grant number OPP1141832].

**Competing interests** None declared.

**Patient and public involvement** Patients and/or the public were not involved in the design, or conduct, or reporting, or dissemination plans of this research.

**Patient consent for publication** Not applicable.

**Ethics approval** This evaluation was approved by the evaluation agency, Population Council's institutional review board. The public was not involved in the design, conduct, reporting, or dissemination plans of this research. Prior to an interview, participants received written information on the survey as well the assurance of confidentiality. Written consent was taken from all interviewees, and interviews were conducted in a private space.

**Provenance and peer review** Not commissioned; externally peer reviewed.

**Data availability statement** Data are available upon reasonable request.

**ORCID iDs**
Laili Irani http://orcid.org/0000-0002-5472-507X
Amnesty Elizabeth LeFevre http://orcid.org/0000-0001-8437-7240

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
