## [Reviewer comments · BMJ Open]

ARTICLE DETAILS

TITLE (PROVISIONAL)	Key learnings from an outcome and embedded process evaluation of a direct to beneficiary mobile health intervention among marginalized women in rural Bihar, India
AUTHORS	Irani, Laili; ., Supriya; Mathur, Ruchika; Verma, Raj Kumar; Mohan, Diwakar; Dhar, Diva; Seth, Aaditeshwar; Chaudhuri, Indrajit; Chaudhury, Mahua; Purthy, Apolo; Nanda, Ankit; Singh, Shivani; Gupta, Akshay; LeFevre, Amnesty

VERSION 1 – REVIEW

REVIEWER	Gonçalves, Carla FCNAUP
REVIEW RETURNED	19-Jul-2021

GENERAL COMMENTS	The work presents relevant findings in the field of health promotion interventions among mothers in poor areas. The paper is very extensive and maybe it could be interesting to have one work with the process analysis and another with the outcomes separately. In introduction the authors should avoid use the first person ("we" page 8, lines 2 and 10).
---

REVIEWER	Khatun, Fatema ICDDR, Health Systems and Population Studies Division
REVIEW RETURNED	23-Dec-2021

GENERAL COMMENTS	This paper was not articulated in a standard format of manuscript. Study methods and qualitative data collection process and analysis was not clear. Please rewrite the paper and submit again . Thank you
---

REVIEWER	Cheng, Christina Swinburne University of Technology
REVIEW RETURNED	26-Feb-2022

GENERAL COMMENTS	Congratulations on a well-written and well-presented manuscript describing the evaluation process of an interactive voice response intervention for marginalised women in rural India. This is an important study to demonstrate lessons learnt from a mobile intervention targeting people experiencing disadvantage and vulnerability. I only have a few comments: 1. Many acronyms were used in the manuscript which can be quite confusing to remember what that you were referring to, can you minimise the use of acronyms for easier reading? 2. How were qualitative analysis conducted? 3. Before scaling up, what about the recommendations to mitigate the barriers identified such as technical issues, financial issues
---

	about using mobile phone, language issue if the program or the cultural issue of expression of apprehension in using mobile phones for long duration in front of male members of the household? How to expand the exposure to the intervention?
--	---

VERSION 1 – AUTHOR RESPONSE

Reviewer: 1

Prof. Carla Gonçalves, FCNAUP, UTAD

Comments to the Author:

The work presents relevant findings in the field of health promotion interventions among mothers in poor areas. The paper is very extensive and maybe it could be interesting to have one work with the process analysis and another with the outcomes separately.

Response to reviewer's comment: Thank you for your feedback. While we appreciate that the paper attempts to compress a lot of relevant information together, some of the key findings from the process evaluation are presented in the manuscript to place the results of the outcome evaluation in context and in order to explain how the intervention was carried out, why some of the intended outcomes of the program did not show improvement and what matters may need to be kept in mind if such a program were to be replicated in a different setting or scaled up to a larger audience. Hence, we request the reviewers to accept these integrated results and review the findings from the embedded process evaluation within the larger context of explaining the results from the outcome evaluation.

In introduction the authors should avoid use the first person ("we" page 8, lines 2 and 10).

Response to reviewer's comment: Thank you for this comment. Use of the first person has been removed from the manuscript.

Reviewer: 2

Dr. Fatema Khatun, ICDDR,B

Comments to the Author:

This paper was not articulated in a standard format of manuscript. Study methods and qualitative data collection process and analysis was not clear. Please rewrite the paper and submit again .

Thank you

Response to reviewer's comment: Thank you for your comment. The manuscript has been rewritten and divided into four main sections: Introduction, Methods, Results and Discussion.

The methods section further describes the evaluation design and conceptual framework, the data collection and sampling process, list of variables analyzed, and statistical procedures utilized for each component of the evaluation, i.e., the outcome evaluation and the three major components of the process evaluation—system readiness, behavioral response and technological context. The process of collecting and analyzing the qualitative data has also been expounded upon each of the sub-sections of the methods section.

The results section presents the results from the outcome evaluation and three elements of the process evaluation.

The discussion section ties all the findings together, presents it in the larger context of existing evidence, discusses the implications of scaling such a program and suggests areas of further research. It highlights some of the limitations of the study and presents a conclusion to the manuscript.

Reviewer: 3

Dr. Christina Cheng, Swinburne University of Technology

Comments to the Author:

Congratulations on a well-written and well-presented manuscript describing the evaluation process of an interactive voice response intervention for marginalised women in rural India. This is an important study to demonstrate lessons learnt from a mobile intervention targeting people experiencing

disadvantage and vulnerability. I only have a few comments:

1. Many acronyms were used in the manuscript which can be quite confusing to remember what that you were referring to, can you minimise the use of acronyms for easier reading?

Response to reviewer: we have removed almost all acronyms from the paper to increase the readability of the paper and make it easier to follow.

2. How were qualitative analysis conducted?

Response to reviewer: We realize that the initial draft of the manuscript did not describe the analysis of the qualitative component of the paper in detail. This has now been expanded upon in the methods section under the various sub-headings, i.e., data collection and sampling, variables and statistical procedures. The qualitative data are explained under the behavioral response of the process evaluation throughout the manuscript.

3. Before scaling up, what about the recommendations to mitigate the barriers identified such as technical issues, financial issues about using mobile phone, language issue if the program or the cultural issue of expression of apprehension in using mobile phones for long duration in front of male members of the household? How to expand the exposure to the intervention?

Response to reviewer: the Discussion section of the manuscript has been revised to include possible responses to some of the questions you have raised above. Further, the sixth paragraph of the Discussion section discusses some of the barriers the program will need to overcome if it wishes to scale up to a larger audience.